# Deep Immune Profiling of Multiple Myeloma at Diagnosis and under Lenalidomide Maintenance Therapy

**DOI:** 10.3390/cancers15092604

**Published:** 2023-05-04

**Authors:** Sini Luoma, Philipp Sergeev, Komal Kumar Javarappa, Tiina J. Öhman, Markku Varjosalo, Marjaana Säily, Pekka Anttila, Marja Sankelo, Anu Partanen, Anne Nihtinen, Caroline A. Heckman, Raija Silvennoinen

**Affiliations:** 1Department of Hematology, Comprehensive Cancer Center, Helsinki University Hospital and University of Helsinki, 00290 Helsinki, Finland; 2Institute for Molecular Medicine Finland, Helsinki Institute of Life Science, iCAN Digital Precision Cancer Medicine Flagship, University of Helsinki, 00290 Helsinki, Finland; 3Institute of Biotechnology, Helsinki Institute of Life Science, University of Helsinki, 00014 Helsinki, Finland; 4Hematology-Oncology Unit, Oulu University Hospital, 90220 Oulu, Finland; 5Hematology Unit, Department of Internal Medicine, Tampere University Hospital, 33520 Tampere, Finland; 6Department of Medicine, Kuopio University Hospital, 70210 Kuopio, Finland; 7Department of Internal Medicine, North Carelia Central Hospital, 80210 Joensuu, Finland

**Keywords:** CyTOF, immunoprofiling, flow cytometry, multiple myeloma, clinical trials, patient stratification

## Abstract

**Simple Summary:**

Currently, multiple myeloma is mostly an incurable disease. Given the poor survival and frequent disease relapses, there is an urgent need for new treatment approaches and patient stratification. In an attempt to predict response to treatment, we comprehensively analyzed the immune profiles of longitudinal bone marrow samples collected from 18 multiple myeloma patients, separating them into good and poor responders to therapy. We identified several distinctive features that might have an influence on the therapy response, such as the abundance of tumor cells and the functional state of T and NK cells. These results suggest that deep immune profiling could be used for the guidance of treatment.

**Abstract:**

The bone marrow microenvironment interacts with malignant cells and regulates cancer survival and immune evasion in multiple myeloma (MM). We investigated the immune profiles of longitudinal bone marrow samples from patients with newly diagnosed MM (*n* = 18) using cytometry by time-of-flight. The results before and during treatment were compared between patients with good (GR, *n* = 11) and bad (BR, *n* = 7) responses to lenalidomide/bortezomib/dexamethasone-based treatment. Before treatment, the GR group had a lower tumor cell burden and a higher number of T cells with a phenotype shifted toward CD8+ T cells expressing markers attributed to cytotoxicity (CD45RA and CD57), a higher abundance of CD8+ terminal effector cells, and a lower abundance of CD8+ naïve T cells. On natural killer (NK) cells, increased expression of CD56 (NCAM), CD57, and CD16 was seen at baseline in the GR group, indicating their maturation and cytotoxic potential. During lenalidomide-based treatment, the GR patients showed an increase in effector memory CD4+ and CD8+ T-cell subsets. These findings support distinct immune patterns in different clinical contexts, suggesting that deep immune profiling could be used for treatment guidance and warrants further exploration.

## 1. Introduction

Interactions between the bone marrow (BM) microenvironment and malignant plasma cells (PCs) are important for treatment outcomes in multiple myeloma (MM) [1]. During the precursor stages of the disease, alterations can already be seen in the BM immune profile [2]. During progression from monoclonal gammopathy of undetermined significance (MGUS) and smoldering MM to active disease, these alterations vary from natural killer (NK) cell abundance in early stages and the loss of granzyme K+ memory cytotoxic T cells in smoldering MM [3] to an abundance of mesenchymal stromal cells with an inflammatory profile in active MM [4]. Malignant PCs evade the immune defense with the help of mesenchymal stromal cells by creating an immunosuppressive microenvironment [5]. This microenvironment is characterized by an increasing number of regulatory T cells and myeloid-derived suppressor cells (MDSCs) [6,7], the dysfunction of antigen presentation [8], and the suppression and exhaustion of effector immune cells [9].

Data demonstrating the impact of the BM immune microenvironment on disease outcomes are still limited. Paiva et al. [10] divided MM patients into three clusters according to their BM immune profiles after treatment. The authors managed to show differences in survival between the clusters, regardless of measurable residual disease (MRD) status. Long-term disease control has been linked to an MGUS-like immune profile [11,12]. The abundance of T and NK cells has been correlated with better outcomes in MM [13,14,15,16,17,18,19]. Parmar et al. [20] analyzed the BM immune profiles of MM patients after autologous stem cell transplantation (ASCT) with cytometry by time-of-flight (CyTOF), and the patient group characterized by more T cells at the opposite ends of the differentiation spectrum (naïve/terminally differentiated) had the worst survival.

Lenalidomide is an immunomodulatory drug with a mechanism based on interaction with cereblon and the ubiquitin ligase complex [21]. It is known to influence T- and NK-cell activity and to enhance antibody-dependent cellular cytotoxicity [22,23].

The FMG-MM02 study by the Finnish Myeloma Group was designed to explore the response to lenalidomide, bortezomib, and dexamethasone (RVD) induction, followed by ASCT and lenalidomide maintenance as a first-line therapy for MM patients (referred to here as “Treatment”) [24,25]. In the present investigation, as a follow-up of the FMG-MM02 study, the BM immune profiles at pre- and post-treatment stages were evaluated by CyTOF to find possible differences between patients with good and bad responses, defined by the length of progression-free survival time (PFS).

## 2. Materials and Methods

### 2.1. Patients and Samples

Eighteen subjects were included in this study. All patients were transplant-eligible and had newly diagnosed MM. The patients received three induction cycles with RVD, followed by stem cell mobilization and a single ASCT. Lenalidomide maintenance treatment with a dose of 10 mg/day on days 1–21 in 28-day cycles was started three months after ASCT and was continued until progression or toxicity. All patients had already been followed for more than 5 years. Based on PFS, patients were divided into 2 response cohorts: the good-response cohort (GR, *n* = 11, PFS > 5 years) and the bad-response cohort (BR, *n* = 7, PFS ≤ 5 years). BM samples were collected in the Finnish Hematology Registry and Clinical Biobank (FHRB Biobank) at several time points during the study: baseline for good (GR, baseline, *n* = 11) and bad responders (BR, baseline, *n* = 7), after the achievement of at least very good partial remission (VGPR) on lenalidomide maintenance (response, *n* = 11), at follow-up if this good response was maintained or improved (long response, *n* = 11), and at relapse on lenalidomide (relapse, *n* = 5) (Figure 1A). Details of the minimal residual disease (MRD) analyses with multiparameter flow cytometry (MFC) assays for the patients who participated in the FMG-MM02 study were presented previously [25]. The median sensitivity of negative samples was <0.01% (<0.004 to <0.02%) for response and <0.0008% (<0.0004 to <0.003%) for long response.

The FMG-MM02 study was approved by the Research Ethics Committee of the Northern Savo Hospital District and the FHRB Biobank Scientific Board, and it was conducted according to the Declaration of Helsinki, International Conference of Harmonization, and Guidelines for Good Clinical Practice and registered at ClinicalTrials.gov with the number NCT01790737. Written informed consent was obtained from all patients before inclusion.

### 2.2. Cell Staining for CyTOF Analysis

Cryopreserved BM mononuclear cells (BMMNCs) were thawed at +37 °C in a water bath, washed with cell staining buffer (CSB: Fluidigm, South San Francisco, CA, USA), centrifuged at 500 RCF for 5 min at room temperature (RT), and then resuspended in CSB. The immune cells were labeled using the Maxpar Direct Immune Profiling Assay Kit, which includes 30 markers (Fluidigm, cat. #201325) (Appendix A). Staining was performed using a modified protocol from the manufacturer. The cells were resuspended at up to 5 × 10^6^ cells per sample in 50 μL of CSB and 5 μL of Human TruStain FcX Fc Receptor Blocking Solution (Biolegend, San Diego, CA, USA, cat. #422301) and then incubated for 10 min at RT. Then, 215 μL of CSB was added to each sample, which was then directly transferred into individual 5 mL tubes containing lyophilized antibodies, mixed well, and then transferred back to the sample tubes and incubated for 30 min at RT in the dark. Staining was followed by fixation in 1 mL of 1.6% paraformaldehyde in PBS. The fixed cells were stored at +4 °C overnight.

### 2.3. Sample Acquisition and Data Normalization

Following overnight incubation, the fixed cells were washed twice in CSB and then pelleted at 1200 RCF for 3 min. The cells were resuspended in 1 mL of Maxpar Fix and Perm buffer containing 0.5 μL of Cell-ID intercalator-Ir191/193 (1:2000) and incubated for 60 min at +4 °C. The cells were washed twice in CSB and twice in cell acquisition solution (CAS) with centrifugation at 1200 RCF for 3 min for each wash. The cells were counted and resuspended at 1 × 10^6^ cells/mL in CAS containing 0.1× EQ Four Element Calibration Beads (Fluidigm, cat. #201078). The cells were filtered into 35 μm mesh cap tubes immediately prior to acquisition.

Sample acquisition was performed on a Helios instrument utilizing software version 6.7.1016 and the Maxpar Direct Immune Profiling Assay template (Fluidigm). The instrument was equipped with a WB injector, and all samples were acquired in CAS containing 0.1× EQ Four Element Calibration Beads (Fluidigm). The instrument was calibrated prior to acquisition. Following the instrument tuning and bead sensitivity test, the system was preconditioned with CAS. A minimum of 2.5 × 10^5^ to 4 × 10^5^ events were acquired per sample at a typical acquisition rate of 250–500 events/s. After acquisition, EQ calibration beads were used to normalize all data within and across experiments. Normalization was performed using CyTOF software version 6.7.1016 (Fluidigm). After that, populations of interest were gated (details in Appendix A).

### 2.4. Computational Analysis

Unsupervised computational methods were applied to visualize high-dimensional mass cytometry data. We applied Uniform Manifold Approximation and Projection (UMAP) [26] to interrogate high-dimensional data as a two-dimensional map. Files for each sample among the studied patient groups were then imported into Cytobank, and then manually gated live cells (see above) were selected to perform UMAP. After optimization, the UMAP algorithm was performed on 30 channels using 30 neighbors, a minimal distance of 0, and a proportional selection of up to 3 million cells in total. In each figure, all samples were derived from the same UMAP run. The FlowSOM algorithm was performed on dimensionally reduced data to cluster the cell populations, with 100 clusters, 20 metaclusters, and 20 iterations. Immune cells were identified based on the distribution and expression of surface markers. Scales on the maps were generated using log1p-transformed, scaled, and centered median values of expression intensity for each marker.

To track shifts in the phenotypes of the manually gated population, we used the RadViz package (v. 0.9.2) for the R environment (v. 4.0.3) with the default settings. For the identification of new populations, CITRUS, SPADE, and FlowSOM algorithms from Cytobank were implemented on the manually gated live cells.

A brief graphic representation of the workflow is shown in Figure 1B.

### 2.5. Statistical Analysis

The samples were cleaned of beads and cells with low quality and viability (Appendix A) and gated, and cell percentages were calculated for every sample using FlowJo software. The percentage of different immune cell populations was exported from FlowJo for further analysis and normalization (Appendix A). Statistical analyses were performed using GraphPad Prism 10 (GraphPad Software, La Jolla, CA, USA) and R software. The Wilcoxon signed-rank test or Mann–Whitney two-tailed U test was used to compare continuous non-parametric variables, and Fisher’s exact test was used for categorical variables; *p* < 0.05 was considered statistically significant. We deliberately omitted the adjustment of the *p*-value, as the low power of our statistical comparisons resulted in non-significant adjusted *p*-values for all comparisons. Therefore, some of the results might have been significant just by pure chance. The following abbreviations were used for *p*-values: *—*p* < 0.05, **—*p* < 0.01, and ***—*p* < 0.001. The probabilities of overall survival (OS) and PFS were calculated with the SPSS 25 (SPSS Inc./IBM, Armonk, NY, USA) software package using the Kaplan–Meier method and the log-rank test for univariate comparisons. OS was defined as the time from treatment initiation until death due to any cause, and PFS was defined as the time from treatment initiation until progression or death due to any cause. The data cut-off was 1 June 2021.

## 3. Results

The baseline characteristics and outcomes of the patients are described in Table 1. There were more patients with high-risk cytogenetic findings in the bad-responder cohort compared to good responders. Detailed cytogenetic findings are described in Appendix A. Patients in the good-responder cohort mainly had IgG-type MM. With a median follow-up time of 83 (range, 13–97) months, the median PFS was not reached in the good-response cohort and was 18 months in the bad-response cohort. Sustained MRD negativity was achieved by 64% in the good-response cohort, compared to none in the bad-response cohort. The best-response samples were collected after reaching VGPR at a median of 24 (range, 9–50) months after ASCT (Appendix A) and after a median of 21 (range, 6–46) months from the beginning of lenalidomide maintenance therapy. Four patients were in VGPR, one was in complete remission (CR), and six were in stringent CR (sCR). Long-good-response samples were collected if at least VGPR was maintained for at least 2 years, at a median of 60 (range, 49–70) months after ASCT, and after a median of 56 (range, 45–67) months of lenalidomide exposure as a maintenance treatment. At that time, two patients were in VGPR, one was in CR, and eight were in sCR. Relapse samples were taken after a median of 6 (range, 2–23) months of lenalidomide maintenance.

### 3.1. Distinct Immune Signatures Were Identified in Good and Bad Responders at Baseline

The unsupervised UMAP algorithm was used for dimensionality reduction, and the FlowSOM algorithm automatically clustered cells based on their projected locations and identified 18 metaclusters, which were divided into major immune cell populations, such as B, T, NK, and dendritic cells, monocytes, and granulocytes (Figure 2A). The phenotypes of the non-malignant clusters are represented in Appendix A, and median raw values of expression intensity are indicated in Appendix A. We noticed the aberrant expression of CD14 on cells not known to express this marker, for example, T cells (Metaclusters 2, 9, and 10). Another important finding was the expression of chemokine receptors on CD4+ T cells (Metacluster 2). We observed on average 2.46 times higher expression of CCR4, CXCR5, and CXCR3 markers on CD4+ lymphocytes in BR, baseline, and relapse groups compared to GR, baseline, and response groups’ values, respectively (Appendix A).

### 3.2. Malignancy-Associated Cell Populations in the Bad-Response Group

A detailed analysis of the immune cells in the BR group identified two malignancy-associated cell (MAC) populations—MAC1 in Metacluster 3 and MAC2 in Metacluster 8 (Figure 2B). These populations were present at baseline and relapse, but not at the stages when responding to myeloma treatment, which let us associate these cells with the malignant state. The MAC1 population was characterized by CD38+NCAM+CD45 −D19-, which is similar to the phenotype of malignant PCs. MAC2 had a mixed phenotype (CD45-/dim CD14+CD38++NCAM+CCR7+CCR6+CXCR4+CXCR3+CXCR5+). MAC1 and MAC2 populations were larger in the BR baseline group (0.50% (0.15–3.70%) and 6.50% (3.50–11.69%), respectively) compared to those in the GR group (0.41% (0.03–0.87%) and 0.71% (0.25–3.84%), respectively; median value and IQR are indicated), although the difference lacked statistical significance. Both populations were successfully eliminated by therapy in good responders (0.0% (0–0.01%) and 0.04% (0.02–0.06%) in the response group, 0.01% (0–0.01%) and 0.04% (0.03–0.07%) in long-response group). In contrast, both populations were detected in higher abundance at relapse (0.07% (0.01–0.1%) for MAC1 and 5.18% (0.43–7.52%) for MAC2) (Appendix A).

### 3.3. Good Responders Have More Cells with an Immunoreactive Phenotype

In the GR baseline group, compared to the BR baseline group, we identified a higher abundance of the total T-cell population (Figure 3A). We also observed that T cells in the GR group tended to express CD8a, CD45RA, and CD57, whereas those in the BR group tended to express CD4, CCR7, and CD45RO. Moreover, deeper analyses showed us differences in the CD8+ T-cell populations in the GR and BR baseline groups. CD8+ T cells in the GR group showed a higher proportion of cells with higher expression of cytotoxicity-associated markers (CD45RA and CD57) and lower expression of markers associated with a more naïve phenotype (CD45RO and CCR7) (Figure 3B), whereas the BR baseline group had fewer CD8+ terminal effector cells and more CD8+ naïve T cells (Figure 3C). Furthermore, in the GR baseline group, we observed higher expression of CD56 (NCAM), CD57, and CD16 on NK cells, indicating both the maturation and cytotoxic potential of these cells (Figure 3D). All together, these findings suggest the presence of a higher number of cytotoxic immune cells and a shift toward an immune effector state in the GR baseline cohort. Interestingly, the analysis of T- and NK-cell exhaustion marker expression did not reveal any dramatic differences between the GR and BR groups (Appendix A). Another difference between the BR and GR baseline groups was the higher numbers of both CD4+ and CD8+ NKT cells in the BR baseline group (Figure 3E). Importantly, in the BR baseline group compared to the similar GR group, there was a higher tumor burden with high expression of classical MM markers, namely, CD38 and CD56 (NCAM), accompanied by the absence of CD45 expression (Figure 3F). Additionally, the majority of patients in the BR group had high-risk cytogenetics (5/7 patients) compared to the GR cohort (3/11 patients). To address the possible influence of cytogenetics on the identified differences, we compared high- and standard-risk cytogenetic groups for the indicated cell populations (Appendix A). These comparisons did not reveal any significant distinctions between the groups.

### 3.4. Longitudinal Analysis of Good Responders at Different Stages

A comparison of longitudinal samples revealed a bias in B-cell populations, showing a shift toward a naïve phenotype at the time of the best response (Figure 4A), while memory B cells increased, and naïve B cells slightly decreased during the course of maintenance therapy (Figure 4B). In contrast, with the best response, both the CD4+ and CD8+ T-cell populations displayed a greater abundance of effector cells and a decrease in naïve cells (Figure 4C,D). In addition, patients with the best response had fewer CD57+ cells, which, despite being highly cytotoxic, are also senescent and not capable of further proliferation. Moreover, the positive change of an increased number of NK cells with the best response was observed (Figure 4E). It is worthwhile to mention that samples in the high-cytogenetic-risk group did not seem to differ from those in the standard-risk group.

Changes in the respective populations between the baseline and relapse stages for bad responders are shown in Appendix A. We observed the same trends for B and CD4+ T cells, which are not, however, statistically significant. As for CD8+ and NK cells, the observed changes are conflicting and do not follow the trends identified in GR cohorts.

## 4. Discussion

This study revealed potential biomarkers associated with MM patients who had a good response to treatment, which consisted of induction RVD, followed by stem cell mobilization, single ASCT, and lenalidomide maintenance. In addition, our observations showed the influence of this treatment on the BM immune profile. Based on the results, we hypothesize that the findings could be used to predict patient response to treatment using assays that would be rapid and minimally invasive.

In the present study, we observed that MM patients who responded to the treatment had higher abundances of effector/cytotoxic cells, including T cells expressing higher levels of CD57 and/or CD45RA and NK cells with CD57, CD16, and CD56 (NCAM), in comparison to patients who did not respond. Additionally, responding patients had a lower tumor burden and displayed the expression of chemokine receptors (CCR7, CCR6, CXCR4, CXCR3, and CXCR5) on CD4+ T cells. Moreover, many immune cell populations of both good and bad responders showed an unusually high expression of CD14. In addition, we identified two populations potentially associated with malignancy—MAC1 and MAC2. Although a classic marker for MM cells, CD138, was absent in our panel, MAC1 demonstrated a classical phenotype of malignant MM cells. At the same time, the MAC2 population, which had a phenotype consistent with malignant PCs, expressed unusual markers specific for monocytes or MDSCs (CD14, chemokine receptors). We hypothesize that the aberrant expression patterns of CD14 and chemokine receptors on different cells might be attributed to the strong interaction between the above-mentioned populations and CD14-expressing monocytes, MDSCs, or their remnants. Another possible explanation is that these aberrant events represent doublets of cells belonging to different lineages.

Furthermore, we showed that GR patients compared to the BR group at baseline had decreased levels of CD4+ and CD8+ NKT cells. Although these observations appear contradictory to our previous findings, previous studies have shown that NKT cells, specifically type II NKT cells, might have an immunosuppressive role [27]. Based on that, we propose that the increased level of NKT cells in the BR group is associated with higher immunosuppression. However, due to the limitations in our panel, we were not able to analyze the subtypes of this population. In addition, we analyzed the immune cell populations according to the cytogenetic risk group. The analysis revealed no significant differences between samples with high- and standard-risk cytogenetic disease. These results support our hypothesis that the observed differences between the shown populations are associated with the treatment response and prognosis.

During the course of the treatment, good responders showed an increase in effector memory CD4+ and CD8+ T-cell subsets, indicating the higher cytotoxic capacity of the immune system, as these populations are known to produce IFNℽ and other factors associated with immunity [28,29,30]. Kalff et al. [19] analyzed peripheral blood immune profiles using CyTOF during lenalidomide maintenance treatment and found similar results in terms of the enrichment of CD4+ and CD8+ T cells in patients with a good response. These consistent findings provide us with tools to better understand the tumor microenvironment and to identify features that characterize a favorable immune profile, which we could use to distinguish patients with a higher chance of achieving a durable response from patients at risk of an early relapse.

When conducting the longitudinal analysis, we observed different behaviors of B and T lymphocytes. Total memory B cells tended to decrease, and naïve B cells simultaneously increased during RVD + ASCT therapy administration, while the initial B-cell levels were restored only on lenalidomide maintenance. The behavior of B cells might be attributed to the action of bortezomib. Pellom et al. [31] showed the increased cytotoxicity of bortezomib toward antibody-secreting B cells, such as memory and PC populations. Subsequently, to restore homeostasis, the maturation of B cells was accelerated, resulting in an increased number of naïve B cells. In contrast, T cells, both CD4+ and CD8+, had the opposite dynamics—an increase in the effector memory population and a decrease in the naïve population. Such behavior confirms our conclusions regarding the favorable immune profiles of patients with a good response in terms of the consistent changes in T and NK cells, leading to the higher intensity of the cytotoxic, effector-like phenotype of the BM immune microenvironment.

A comparison of bad responders at different stages showed that the B-cell and CD4+ T-cell populations of interest seemed to follow the same trend as shown for cohorts of good responders. However, the CD8+ T-cell population dynamics tended to be different in the BR and GR cohorts, and this might lead to their response-defining role, having an additive or synergistic effect with the above-mentioned changes caused by the treatment in general for other lymphocyte populations.

Our study has some limitations. First, the patient number in the study cohorts was small. There were considerable differences in the disease characteristics between the good- and bad-responder cohorts, with the good-response cohort having fewer patients with high-risk disease, reflecting the better prognosis of this cohort. Second, we did not have markers to distinguish small cell subpopulations, such as different NKT cell types, since the panel used was designed for the identification of the major immune cell populations. ASCT may have had a major influence on the immune cell populations in the first good-response samples, although they were all taken 9 months after ASCT at the earliest. Nevertheless, this pilot project provides novel data regarding deep immunoprofiling correlating with response to treatment.

## 5. Conclusions

In summary, we suggest that the assessment of the immune system status (cytotoxic/effector features of T and NK cells and the expression of chemokine receptors on CD4+ T cells) along with the tumor burden might be a novel approach to predict response to treatment. Moreover, the identified abnormal expression of chemokine receptors on CD4+ lymphocytes in bad responders could be used for the early recognition of bad responders, either directly (by measuring the expression of chemokine receptors) or indirectly (by assessing the levels of chemokines in peripheral blood). Randomized clinical trials with larger cohort sizes are warranted to verify these findings regarding guidance to assist treatment and the early identification of patients’ potential response with less invasive tests using peripheral blood.

## Figures and Tables

**Figure 1 cancers-15-02604-f001:**
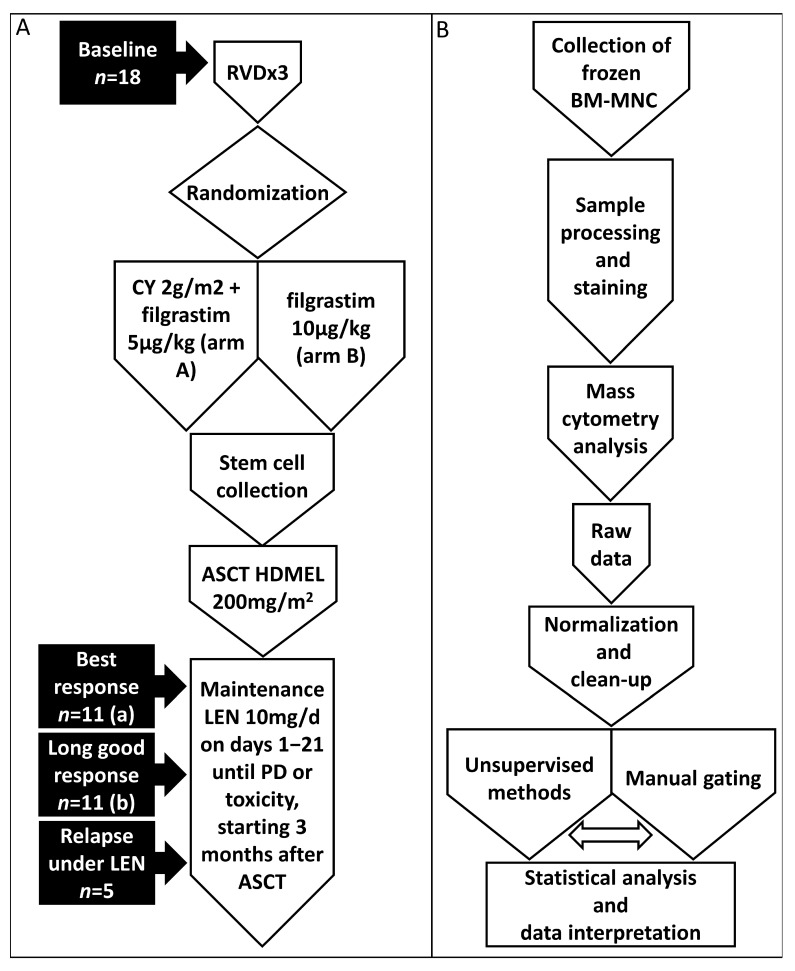
Workflows of clinical and immune profiling studies. (**A**) Treatment and sample collection. (**B**) Schematic representation of the immune profiling study workflow. Abbreviations: RVD, lenalidomide/bortezomib/dexamethasone; CY, cyclophosphamide; ASCT, autologous stem cell transplantation; HDMEL, melphalan 200 mg/m^2^; LEN, lenalidomide; PD, progressive disease. (a) Four patients were in VGPR, one was in CR, and six were in sCR. (b) Three samples were taken after discontinuation of LEN maintenance.

**Figure 2 cancers-15-02604-f002:**
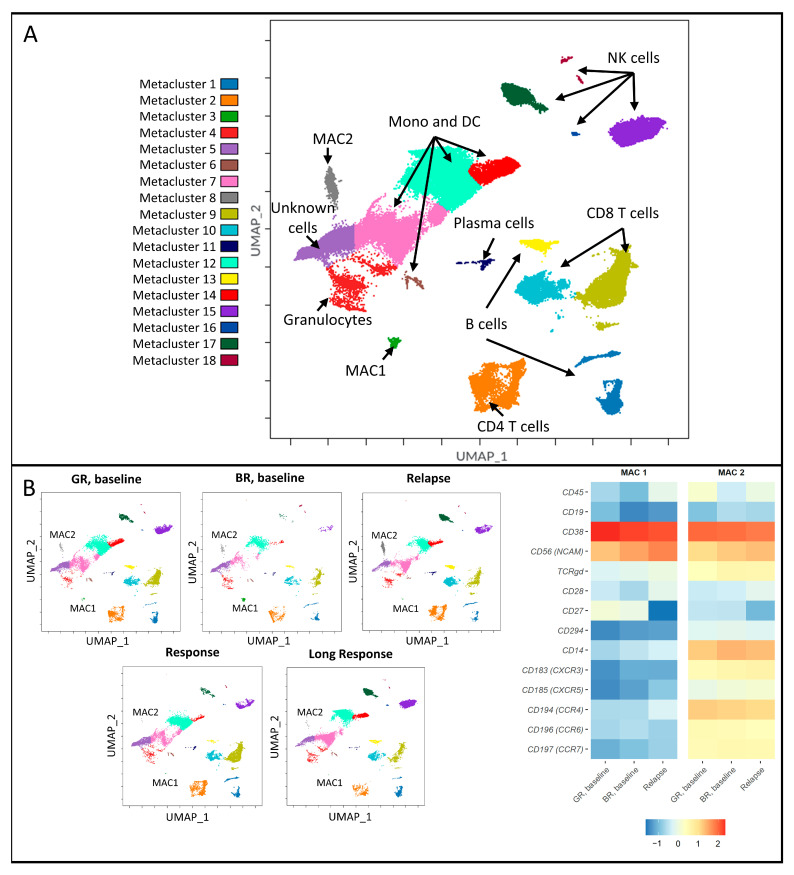
Overview of bone marrow composition and phenotypes of the main immune cell lineages. (**A**) Main metaclusters identified by FlowSOM algorithm. Different metaclusters of cells are represented by the respective colors and were overlaid using the same model of dimensionality reduction. (**B**) UMAP plots showing the dynamics of malignancy-associated cells before and after treatment and at relapse. On the right side, the heatmap with the phenotypes of those populations is shown. Markers with higher intensity were manually selected. For the heatmap, log1p-transformed, scaled, and centered median values of expression intensity were used.

**Figure 3 cancers-15-02604-f003:**
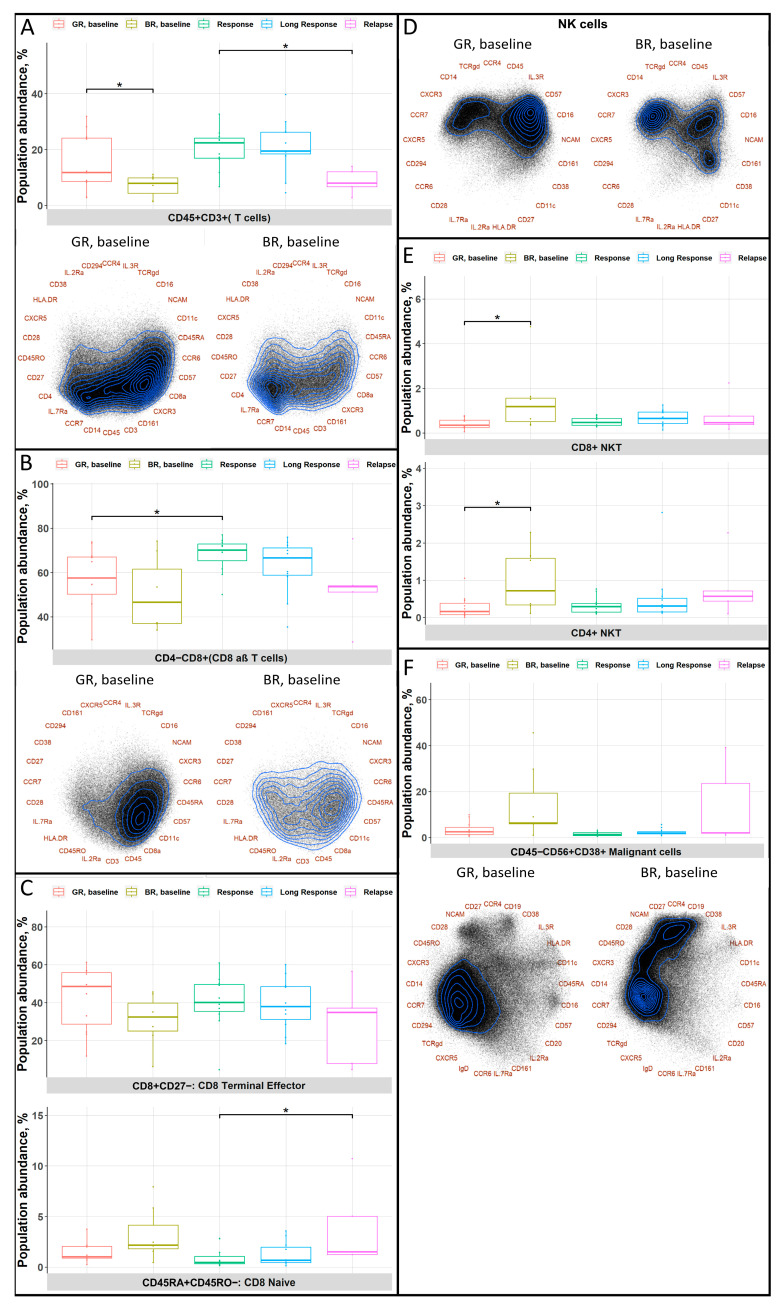
Comparative analysis of immune cell differences between good and bad responders at baseline. (**A**–**F**) The median abundances with 25th and 75th percentiles and whiskers of populations in different patient cohorts and/or Radviz projections with contour plots, representing the density of cells with regard to the expression of markers, are shown. Statistically significant changes are marked with *. The following abbreviations were used for *p*-values: *—*p* < 0.05.

**Figure 4 cancers-15-02604-f004:**
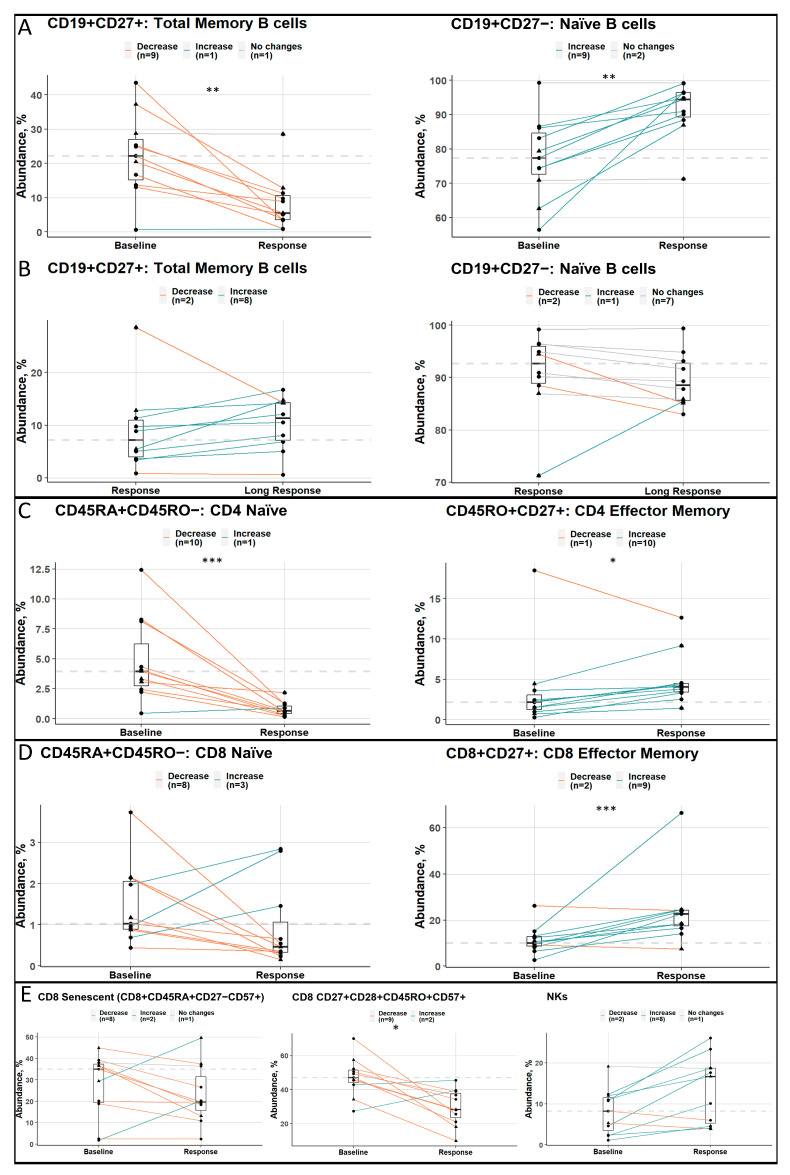
Longitudinal analysis of good responders at different stages. (**A**–**E**) Boxplots with median values, hinges, and whiskers are presented. Values for the same patient are linked by the line. If the ratio between two groups was in a range from 0.9 to 1.1, it was depicted as “No changes”. The dashed line represents the median value of the populations’ abundances at the earlier time point on the plot. Triangles indicate samples with high cytogenetic risk, and circles indicate standard cytogenetic risk. Statistically significant changes are marked. The following abbreviations were used for *p*-values: *—*p* < 0.05, **—*p* < 0.01, and ***—*p* < 0.001.

**Table 1 cancers-15-02604-t001:** Patient characteristics, survival outcomes, and MRD status. (a) t (4;14), t (14;16), t (14;20), del17p13 (2 patients, 82–85%), or gain 1q by FISH. (b) Long-response samples were obtained only from the good-response cohort. (c) MRD-negative by flow cytometry ≥ 1 year during the study. Abbreviations: R-ISS, Revised International Staging System; IMWG risk, International Myeloma Working Group Risk Stratification; PFS progression-free survival; OS overall survival; MRD, minimal residual disease; NR, not reached.

Characteristics	Patients with Good Response	Patients with Bad Response	*p*
*N* (%)	11 (61)	7 (39)	
Age at treatment start, years, median (range)	61.5 (52.9–67.2)	61.0 (56.0–70.0)	0.536
Gender male/female, *n* (%)	10/1 (91/9)	5/2 (71/29)	0.528
Myeloma subtype, *n* (%)			0.050
IgG kappa	6 (55)	2 (29)	
IgG lambda	4 (36)	0 (0)	
IgA kappa	1 (9)	2 (29)	
IgA lambda	0 (0)	1 (14)	
Kappa light chain	0 (0)	1 (14)	
Lambda light chain	0 (0)	1 (14)	
R-ISS stage, *n* (%)			0.067
I	5 (45)	0 (0)	
II	6 (55)	6 (86)	
III	0 (0)	1 (14)	
IMWG risk, *n* (%)			0.748
Low	2 (18)	0 (0)	
Standard	8 (73)	6 (86)	
High	1 (9)	1 (14)	
Cytogenetic risk, *n* (%)			0.145
Standard	8 (73)	2 (29)	
High(a)	3 (27)	5 (71)	
Median duration of lenalidomide maintenance, months, (range)	72.9 (43.7–90.2)	9.9 (1.5–33.9)	<0.001
PFS, median, months, (range)	NR (74.2–96.8)	17.7 (8.7–57.5)	<0.001
OS, median, months, (range)	NR (76.1–96.8)	34.5 (12.5–86.9)	<0.001
MRD-negative at response, *n* (%)			1.000
Yes	5 (50)	1 (100)	
No	1 (10)	0 (0)	
Unknown	4 (40)	0 (0)	
MRD-negative at long response(b), *n* (%)			
Yes	6 (55)		
No	4 (36)		
Unknown	1 (9)		
Sustained MRD-negative (c), *n* (%)			<0.001
Yes	7 (64)	0 (0)	
No	0	6 (86)	
Unknown	4 (36)	1 (14)	

## Data Availability

The datasets generated and analyzed during the study are available from the corresponding author on reasonable request.

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
