# Peer review of "Deep Immune Profiling of Multiple Myeloma at Diagnosis and under Lenalidomide Maintenance Therapy"

_cancers, 2023, doi:10.3390/cancers15092604_

Round 1
Reviewer 1 Report
Following up the immune profiling of myeloma patients is very important issue. The authors have shown several immunological features of good or bad responders, including clear correlation of increase number of functional CD8 T cells in good responders. However, the points below should be corrected.
1. Timepoint of sample collection of each patients should be provided.
2. In figure 4, only data of good responders is shown. Longitudinal analysis of bad responders should be provided as supplemental figures.
3. The comparison between BM sample and PB sample at the same point may provide more information.
4. The text size in each figure is too small.
Reviewer 2 Report
Luoma et al. report on deep immune profiling in longitudinal samples of multiple myeloma patients that underwent RVD induction therapy. The manuscript is well-written. The role of different subtypes of immune cells are of interest, in particular regarding long-term PFS in the context of lenalidomide therapy. The strengths of the manuscript are the use of an up-to-date triplet induction therapy and the longitudinal approach. Weaknesses are present regarding the main confounder, namely the presence of high-risk cytogenetics in the bad responder group in 71% of patients versus 27% of patients in the good responder group. Therefore, the claim that deep immune profiling can guide treatment decisions might not be fully supported by the data. I suggest that the authors highlight this weakness and include a further analysis of immune cell abundance in high risk cytogenetics versus low risk cytogenetics.
MAJOR:
Please show the differences in immune cell abundance (as shown in Figure 3) grouped by high-risk versus standard risk cytogenetics (e.g., as supplementary figures) and describe this analysis in the results section
Table 1: Univariate analysis regarding main parameters should be provided
Figure 3: Please restructure the figure, e.g., by using common.legend in R in order to highlight the contour plot. Boxplots could be reduced in size
MINOR:
Table 1: Light chain subtype should be provided
Figures: The overall quality of figures is well above average. However, the labelling is too small in most of the panels. Please enlarge.
Please correct typos and punctuation, e.g., line 349, 'Treatment'
Please provide the detailed cytogenetics for all patients as supplementary table
Round 2
Reviewer 2 Report
The authors adequately and thoroughly adressed my comments. I suggest acceptance of the manuscript.